# Outcomes of Kidney Transplant Patients with Atypical Hemolytic Uremic Syndrome Treated with Eculizumab: A Systematic Review and Meta-Analysis

**DOI:** 10.3390/jcm8070919

**Published:** 2019-06-27

**Authors:** Maria L. Gonzalez Suarez, Charat Thongprayoon, Michael A. Mao, Napat Leeaphorn, Tarun Bathini, Wisit Cheungpasitporn

**Affiliations:** 1Division of Nephrology, Department of Medicine, University of Mississippi Medical Center, MS 39216, USA; 2Division of Nephrology and Hypertension, Mayo Clinic, Rochester, MN 55905, USA; 3Division of Nephrology and Hypertension, Mayo Clinic, Jacksonville, FL 32224, USA; 4Department of Nephrology, Department of Medicine, Saint Luke’s Health System, Kansas City, MO 64111, USA; 5Department of Internal Medicine, University of Arizona, Tucson, AZ 85721, USA

**Keywords:** atypical hemolytic uremic syndrome, eculizumab, kidney transplantation, renal transplantation, meta-analysis

## Abstract

Background: Kidney transplantation in patients with atypical hemolytic uremic syndrome (aHUS) is frequently complicated by recurrence, resulting in thrombotic microangiopathy in the renal allograft and graft loss. We aimed to assess the use of eculizumab in the prevention and treatment of aHUS recurrence after kidney transplantation. Methods: Databases (MEDLINE, EMBASE and Cochrane Database) were searched through February 2019. Studies that reported outcomes of adult kidney transplant recipients with aHUS treated with eculizumab were included. Estimated incidence rates from the individual studies were extracted and combined using random-effects, generic inverse variance method of DerSimonian and Laird. Protocol for this systematic review has been registered with PROSPERO (International Prospective Register of Systematic Reviews; no. CRD42018089438). Results: Eighteen studies (13 cohort studies and five case series) consisting of 380 adult kidney transplant patients with aHUS who received eculizumab for prevention and treatment of post-transplant aHUS recurrence were included in the analysis. Among patients who received prophylactic eculizumab, the pooled estimated incidence rates of recurrent thrombotic microangiopathy (TMA) after transplantation and allograft loss due to TMA were 6.3% (95%CI: 2.8–13.4%, *I*^2^ = 0%) and 5.5% (95%CI: 2.9–10.0%, *I*^2^ = 0%), respectively. Among those who received eculizumab for treatment of post-transplant aHUS recurrence, the pooled estimated rates of allograft loss due to TMA was 22.5% (95%CI: 13.6–34.8%, *I*^2^ = 6%). When the meta-analysis was restricted to only cohort studies with data on genetic mutations associated with aHUS, the pooled estimated incidence of allograft loss due to TMA was 22.6% (95%CI: 13.2–36.0%, *I*^2^ = 10%). We found no significant publication bias assessed by the funnel plots and Egger’s regression asymmetry test (*p* > 0.05 for all analyses). Conclusions: This study summarizes the outcomes observed with use of eculizumab for prevention and treatment of aHUS recurrence in kidney transplantation. Our results suggest a possible role for anti-C5 antibody therapy in the prevention and management of recurrent aHUS.

## 1. Introduction

Atypical hemolytic uremic syndrome (aHUS) is a microvascular occlusive disorder characterized by hemolytic anemia, thrombocytopenia and acute kidney injury that is not associated with Shiga toxin-producing Escherichia coli (STEC) or ADAMTS13 deficiency. Instead, it is typically associated with dysregulation of the alternative complement pathway [1,2]. Thrombotic microangiopathy (TMA) is the pathological lesion seen with aHUS, which represents a response to endothelial injury [3]. About 10% of hemolytic uremic syndrome (HUS) pediatric cases and the majority of cases in adults are due to atypical HUS [4]. 

Kidney transplantation in patients with aHUS has been linked to poor outcomes due to high recurrence rate and graft failure. Approximately 50% of patients with aHUS develop end stage renal disease (ESRD) with a high risk of recurrence after kidney transplantation [5]. Roughly 60–70% of aHUS patients have mutations in factors of the complement system or antibodies directed against complement factor H (*CFH*) [6]. In some cases, aHUS recurrence is noted early after transplantation, while other cases may be at lower risk of recurrence [7]. The risk of aHUS recurrence after transplant is higher in patients with gene mutations that encode circulating complement factors (*C3, CFH,* complement factor *I* (*CFI*)) when compared to patients with gene mutations that encode solid phase proteins such as CD46 [6,8,9]. Patients with no prior history of aHUS may also present with de novo aHUS after kidney transplantation. 

Therapies described for management of aHUS in the post-transplant period include the use of plasma exchange (PLEX), rituximab (used for anti-Factor *H* autoantibodies; helps to maintain low levels of antibodies, preventing recurrence of aHUS after transplant) [10], simultaneous liver–kidney transplant for *CFH* mutations, and use of eculizumab (a humanized monoclonal antibody directed against complement protein C5 and thus inhibits terminal complement activation) [11,12,13]. Prior to the use of eculizumab, patients with gene mutations *CFH*, *CFH*-*CFHR1/3*, *CFI*, *C3*, and *CFB* had a 50% risk of progression to ESRD or death at onset of recurrent aHUS during the first year, and this risk increased to 75% after 3–5 years [14]. 

The KDIGO workgroup recommends the prophylactic use of eculizumab in kidney transplant patients at high risk of recurrence based on their genetic mutations [14]. Whether there is an advantage of preemptive use of eculizumab in all patients with a known pretransplant history of aHUS is currently unclear. In addition, eculizumab use is associated with an increased risk of infection due to terminal complement blockade such as meningococcal infections [15,16]. In this study, we aimed to assess the use of eculizumab in the prevention and treatment of aHUS recurrence after kidney transplantation.

## 2. Methods

### 2.1. Search Strategy and Literature Review 

The protocol for the systematic review has been registered in PROSPERO (registration number: CRD42018089438; http://www.crd.york.ac.uk/PROSPERO). A systematic literature review of EMBASE (1988 to February 2019), MEDLINE (1946 to February 2019), and the Cochrane Database of Systematic Reviews (CDSR) (database inception to February 2019) was performed to assess the use of eculizumab in the prevention and treatment of aHUS recurrence after kidney transplantation. The systematic literature search was undertaken independently by two investigators (M.G.S. and C.T.) using a search approach that incorporated the terms of “kidney” OR “renal” AND “transplant" OR “transplantation” AND “eculizumab”. The search strategy is provided in online Appendix A. No language restriction was applied. A manual literature search for conceivably pertinent studies using references of the included articles was also performed. This study was conducted by the PRISMA (Preferred Reporting Items for Systematic Reviews and Meta-Analysis) statement [17].

### 2.2. Selection Criteria

Eligible studies must be (1) clinical trials or observational studies (cohort, case-series, or cross-sectional studies) that reported use of eculizumab in the prevention and treatment of aHUS recurrence after kidney transplantation; (2) adult (age ≥ 18 years old) kidney transplant recipients; and (3) they must provide the data on outcomes of interest including rates of aHUS recurrence and allograft loss among patients who received prophylactic eculizumab and rates of allograft loss among patients who received eculizumab for treatment of post-transplant aHUS recurrence. The eculizumab treatment group included post-transplant patients with de novo or recurrent aHUS. We excluded case reports and studies with single cases treated with eculizumab. Retrieved studies were independently reviewed for eligibility by the two authors (M.L.G.S. and C.T.). Discrepancies were discussed and resolved by a third author (W.C.) and common consensus. Inclusion was not confined by the size of study. Newcastle-Ottawa quality assessment scale [18] was used to appraise the quality of observational studies and the Cochrane risk-of-bias tool correspondingly for clinical trials [19]. Detailed evaluation of each study is presented in online Appendix A.

### 2.3. Data Abstraction

A structured information collecting form was used to obtain the following information from each study including title, name of the first author, publication year, country where the study was conducted, demographic data of kidney transplant patients, history of previous kidney transplantation, type of donor, genetic mutations associated with aHUS, eculizumab regimen, use of PLEX, and outcomes following kidney transplantation (rates of aHUS recurrence and allograft loss among patients who received prophylactic eculizumab and rates of allograft loss among patients who received eculizumab treatment for post-transplant aHUS recurrence).

### 2.4. Statistical Analysis

Analyses were conducted utilizing the Comprehensive Meta-Analysis 3.3 software (version 3; Biostat Inc., Englewood, NJ, USA). We estimated the pooled incidence rate of recurrent TMA, all-cause allograft loss, and allograft loss due to TMA by the generic inverse variance approach of DerSimonian and Laird, which designated the weight of each study based on its variance [20]. Due to the possibility of between-study variance, we used a random-effects model rather than a fixed-effect model. Sensitivity analysis was performed with restriction to only cohort studies with data on genetic mutations associated with aHUS. We used Cochran’s Q test and *I*^2^ statistic to assess the between-study heterogeneity. A value of *I*^2^ of 0% to 25% represents insignificant heterogeneity, 26% to 50% low heterogeneity, 51% to 75% moderate heterogeneity and 76–100% high heterogeneity [21]. The presence of publication bias was assessed by both subjective inspections of funnel plot and the Egger test [22]. The raw data for this systematic review is publicly available through the Open Science Framework (URL: osf.io/2pz4k).

## 3. Results

A total of 1096 potentially eligible articles were identified using our search strategy. After the exclusion of 888 articles based on title and abstract for clearly not fulfilling inclusion criteria on the basis of type of article, study design, or population or outcome of interest, and exclusion of 172 articles due to being duplicates; 36 full-length articles were reviewed. Fifteen of these articles were subsequently excluded from full-length review because they were also duplicates [23,24,25,26,27,28], did not fulfill inclusion criteria [29,30,31,32,33], did not report the outcomes of interest, or data was unable to be abstracted [34,35,36,37]. An additional three articles were excluded because they were case reports or single cases treated with eculizumab [8,38,39]. Ultimately 18 studies (13 cohort studies and five case series) [5,9,15,40,41,42,43,44,45,46,47,48,49,50,51,52,53,54] consisting of 380 patients with a history of aHUS who received eculizumab for prevention and treatment of post-transplant recurrent aHUS were included into the final analysis. The literature review and selection process are shown in Figure 1. The characteristics of the included studies are presented in Table 1 and Table 2. 

The baseline characteristics of kidney transplant patients and data on genetic mutations associated with aHUS are summarized in Table 3. There were 192 patients in the treatment group and 188 patients in the prophylaxis group, with a median age of 38 and 32.3 years, respectively. Females were predominant in both groups (59.4% vs. 84.4%, respectively). The most commonly reported genetic mutation in this our study population was CFH, followed by CFI and C3 in both groups.

A history of aHUS prior to kidney transplantation was known in all patients included in this systematic review, except for de novo cases. The triggers for post-transplant aHUS were not reported in the majority of patients. The most commonly reported potential triggers were: Viral infections, urosepsis, *C. difficile* infection, tacrolimus use, and everolimus use. De novo aHUS was reported in 56 patients [15,42,47,49]. Tacrolimus was identified as a possible trigger in one of the de novo cases in which aHUS presented 16 years after transplant without any genetic mutations identified [49], while it was suspected in four other cases that had antibody-mediated rejection and their immunosuppression was switched from tacrolimus to sirolimus [42,49]. Urosepsis was reported as a trigger in one de novo case [47].

### 3.1. Use of Eculizumab in the Prevention of aHUS Recurrence after Kidney Transplantation

Data on the initiation of prophylactic eculizumab therapy are summarized in Table 3. Overall, among patients who received prophylactic eculizumab, the pooled estimated rates of aHUS recurrence and allograft loss due to TMA were 6.3% (95%CI: 2.8–13.4%, *I*^2^ = 0%, Figure 2A) and 5.5% (95%CI: 2.9–10.0%, *I*^2^ = 0%, Figure 2B), respectively. 

Sensitivity analysis excluding studies with potentially duplicated patients was performed, and showed the pooled estimated rates of aHUS recurrence and allograft loss due to TMA of 5.5% (95%CI: 1.9–14.6%, *I*^2^ = 0%, Appendix A) and 5.3% (95%CI: 2.4–11.3%, *I*^2^ = 0%, Appendix A), respectively. When analysis was limited only to studies with a mean follow-up time >12 month (mean follow up time range 21 to 35 months), we found pooled estimated rates of aHUS recurrence and allograft loss due to TMA of 4.6% (95%CI: 1.5–13.3%, *I*^2^ = 0%) and 4.8% (95%CI: 2.1–10.7%, *I*^2^ = 0%), respectively.

### 3.2. Use of Therapeutic Eculizumab for aHUS Recurrence after Kidney Transplantation

Data on the initiation of eculizumab treatment for post-transplant aHUS recurrence are summarized in Table 3. Among those who received eculizumab for treatment of post-transplant aHUS recurrence, the pooled estimated rates of allograft loss due to all causes was 24.4% (95%CI: 15.9–35.6%, *I*^2^ = 23%, Figure 3A). The pooled estimated rates of allograft loss due to TMA was 22.5% (95%CI: 13.6–34.8%, *I*^2^ = 6%, Figure 3B). Meta-regression analysis demonstrated no significant correlations between concomitant use of PLEX and rates of allograft loss due to all causes or due to TMA (*p* = 0.61 and 0.18, respectively).

Sensitivity analysis excluding studies that potentially included duplicate patients was performed, and showed the pooled estimated rates of allograft loss due to all causes was 26.7% (95%CI: 13.0–46.9%, *I*^2^ = 33%, Appendix A) and the pooled estimated rates of allograft loss due to TMA was 20.0% (95%CI: 9.1–38.4%, *I*^2^ = 29%, Appendix A), respectively. Sensitivity analysis restricted only to cohort studies with data on genetic mutations associated with aHUS was performed. The pooled estimated incidence of allograft loss due to TMA in cohort studies with data on genetic mutations was 22.6% (95%CI: 13.2–36.0%, *I*^2^ = 10% Figure 4). 

Subgroup analysis stratified by mean follow-up time was performed. We found the pooled estimated rates of allograft loss due to all causes was 25.5% (95%CI: 14.8–40.3%, *I*^2^ = 34%) at a mean follow-up time of 12 to 24 months and 26.0% (95%CI: 3.9–75.1%, *I*^2^ = 13%) at a mean follow-up time of 60 to 72 months. The pooled estimated rates of allograft loss due to TMA were 22.6% (95%CI: 12.1–38.1%, *I*^2^ = 8%) at a mean follow-up time of 12 to 24 months and 26.0% (95%CI: 3.9–75.1%, *I*^2^ = 13%) at a mean follow-up time of 60 to 72 months.

### 3.3. Evaluation for Publication Bias

We found no significant publication bias as assessed by the funnel plots (Figure 5) and Egger’s regression asymmetry test for the rates of aHUS recurrence and allograft loss due to TMA in patients treated with prophylactic eculizumab therapy (*p* = 0.48 and 0.28, respectively), nor for rates of allograft loss due to all causes and allograft loss due to TMA in patients treated with therapeutic eculizumab for aHUS recurrence after kidney transplantation (*p* = 0.78 and 0.20, respectively).

## 4. Discussion

Atypical HUS recurrence after transplant is diagnosed with the presence of laboratory abnormalities such as renal failure, microangiopathic hemolytic anemia, thrombocytopenia and microvascular occlusion [1,48]. In many cases, aHUS recurrence is confirmed with a kidney biopsy–showing the usual findings of glomerular intracapillary thrombosis, congestion, endothelial swelling, and thickening of the capillary wall [4]. 

Our systematic review showed that mutations to genes encoding *CFH*, followed by mutations in *CFI* genes were the most commonly reported in patients with aHUS recurrence after transplant. This observation, however, is limited to the studies included in our analysis, which only included kidney transplant patients treated with eculizumab and did not include patients treated with alternative therapies. In our systematic review, mutations in *CFH* and *CFH/CFHR1* were more commonly associated with graft loss in patients who had received eculizumab therapy. The current literature describes the presence of complement factor genetic mutations in approximately 30% of patients who present with de novo HUS after kidney transplantation [6]. However, no mutations were identified in four patients who presented with de novo aHUS and had graft loss [47,49]. 

We found that in most pretransplant patients with a history of aHUS who had received eculizumab prophylaxis, this was started on the day of kidney transplant surgery (data abstracted from Mallet et al. [47] and Manani et al. [49] were not utilized for this subgroup analysis as only one case in each publication was prophylactically managed with eculizumab). Less than 6% of the patients who received eculizumab prophylaxis presented with aHUS recurrence, and recurrence was more likely to occur when eculizumab was discontinued. Yelken et al. reported their clinical experience in a retrospective cohort of seven patients with a prior history of aHUS who had received eculizumab prophylaxis and underwent kidney transplantation. None of them had presented with aHUS recurrence or graft loss during the follow-up period of 28 months [52]. More recently, a global registry for aHUS enrolled 1549 patients over a 5-year period. One hundred and eighty-eight patients underwent kidney transplantation. From those, 88 patients received eculizumab before or during transplant surgery. This group of patients presented significantly better graft function when compared to those patients who received eculizumab with aHUS recurrence or de novo aHUS in the post-transplant period [15]. 

The time for recurrence of aHUS after transplant in the eculizumab therapeutic group varied, with recurrence seen as early as 3 days and up to 6 years (median of 2 months). Our study shows that allograft outcomes of patients treated after early recurrence (3 days to 3 months post-transplant) [25,28,42,44,49] when compared to those patients treated for late recurrence (29–96 months post-transplant) [15,47,50] were similar. The effect of timing for therapeutic eculizumab is also unclear. Some studies have concluded that early therapy with eculizumab for aHUS recurrence has failed to show better allograft survival [9,47,48]. Future prospective randomized control trials would help elucidate whether early initiation of therapy has a beneficial effect for allograft survival. 

In many cases, management for aHUS recurrence in kidney transplantation includes the use of PLEX. Previously, Le Quintrec et al. had reported a trend towards decreased aHUS recurrence in patients treated preemptively with PLEX [44]. Zuber et al.’s retrospective cohort had subsequently compared PLEX versus eculizumab in both therapeutic and prophylactic modalities. They found that eculizumab was superior when compared to PLEX in preventing and treating aHUS recurrence after kidney transplantation [51]. Other studies have shown no difference in graft survival independently of whether patients received PLEX or not [44,51]. More recently, Favi et al. reported a case-control study where patients with and without PLEX were compared with eculizumab prophylaxis alone. This is the only study that we encountered where a comparison group was used. Patients who did not receive eculizumab (whether they received prophylaxis with PLEX or not) were more likely to have higher graft loss recurrence and allograft rejection when compared to patients who had received eculizumab [54]. Our systematic review showed that 36.2% of individuals required eculizumab after PLEX therapy, while 23.1% received concomitant PLEX in the prophylaxis group. There is no data available regarding the use of PLEX in the rest of the cases. Those patients who received PLEX plus eculizumab had no significant difference in allograft survival and aHUS recurrence in comparison to those without PLEX, (*p* = 0.65). However, it remains difficult to conclude about the effects of concomitant PLEX and eculizumab use in aHUS post-transplant recurrence and associated graft loss due to the small number of patients analyzed, and these patients of interest may have had more severe risk markers that prompted dual therapy. PLEX is still considered an option in the management for aHUS recurrence [15].

Duration of eculizumab therapy has been controversial and so far is based on expert opinion, as there is no strong evidence to support lifelong therapy. We found that most patients who were prophylactically treated had continued eculizumab up to the time of their respective studies’ publication. There is a report of one patient in whom no complement mutation was identified that stopped prophylactic eculizumab after 28.7 months. No aHUS recurrence was identified after 9 months of follow up [46]. The median duration of therapy was 18.9 months in the therapy patient group and 21 months in the prophylaxis patient group. While aHUS allograft recurrence after eculizumab cessation was reported in 5.2% of patients, it is difficult to conclude if other recurrent aHUS cases were missed due to the length of follow up after eculizumab therapy discontinuation. 

Acute allograft rejection rates were similar in treatment and prophylactic groups. Despite receiving eculizumab treatment for aHUS post-transplant recurrence, graft loss was seen in 20.3% of the patients compared to 7.9% of patients who received eculizumab prophylaxis. While graft loss in the treatment group was likely associated with recurrence of aHUS, in the prophylaxis group, two cases of kidney allograft loss were associated with the presence of intestinal hemorrhage (one case immediately after transplant [45], the other case presented four months after transplant [47]). Whether there was a possible association of acute arterial allograft thrombosis with aHUS remains unclear. 

We acknowledge the limitations inherent in this study. Firstly, this systematic review included only observational studies (cohorts and case series). Consequently, the majority of available studies also lacked a comparison (control) group. Only one of the included studies had a comparison group who did not receive eculizumab [54]. Recently, Duineveld C. et al. described 17 patients with aHUS who underwent living donor kidney transplantation without prophylactic eculizumab with median follow-up of 25 months after kidney transplantation [8]. Their institution’s protocol emphasized lower target tacrolimus level and blood pressure control among these patients with aHUS. They found only one patient developed aHUS recurrence at 68 days after kidney transplantation, and was treated successfully with eculizumab. The investigators suggested that living donor kidney transplantation in aHUS without prophylactic eculizumab treatment appears feasible [8]. Secondly, inconsistencies in the reporting of certain variables such as type of donor, history of previous transplantation, genetic mutations, and immunosuppression regimen can make it difficult to draw firm associations between eculizumab prophylaxis or treatment on renal graft outcomes. Caution should be exercised in interpreting these data. Lastly, this study analysis was conducted on a highly selected study population. Only studies in which eculizumab was used were analyzed, and individuals who received an alternate treatment course or did not receive eculizumab were excluded from our analysis. Therefore, the frequency of mutations might not be representative. Although funnel plots and Egger’s test of event rates demonstrated no statistical significance, a low risk of publication bias cannot be implied given the lack of comparison groups. Thus, future studies to assess the responsiveness to eculizumab based on certain aHUS genetic mutations (low risk vs. moderate risk vs. high risk of recurrence) [6] are needed. 

In summary, our study describes the outcomes observed with use of eculizumab for prevention and treatment of aHUS recurrence in kidney transplantation. Our results suggest a possible advantageous role for anti-C5 antibody therapy in the prevention and management of recurrent aHUS. Future prospective studies and clinical trials are needed to evaluate for the efficacy of eculizumab, timing of initiation, and duration of prophylaxis and treatment therapy according to genetic mutations. 

## Figures and Tables

**Figure 1 jcm-08-00919-f001:**
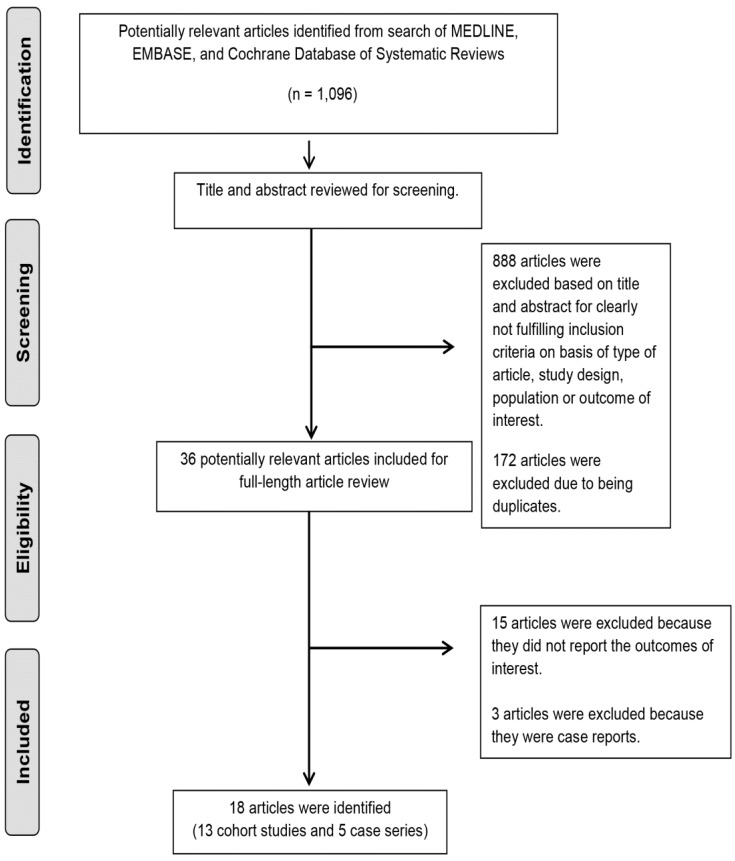
Preferred Reporting Items for Systematic Reviews and Meta-Analysis (PRISMA) flow diagram.

**Figure 2 jcm-08-00919-f002:**
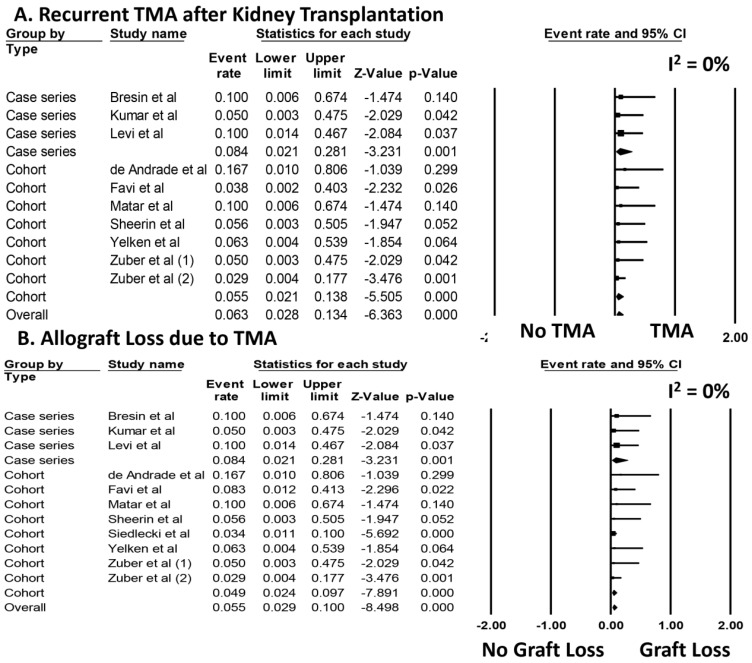
Incidence of (**A**) aHUS recurrence (recurrent TMA) and (**B**) allograft loss due to TMA after kidney transplantation with prophylactic eculizumab.

**Figure 3 jcm-08-00919-f003:**
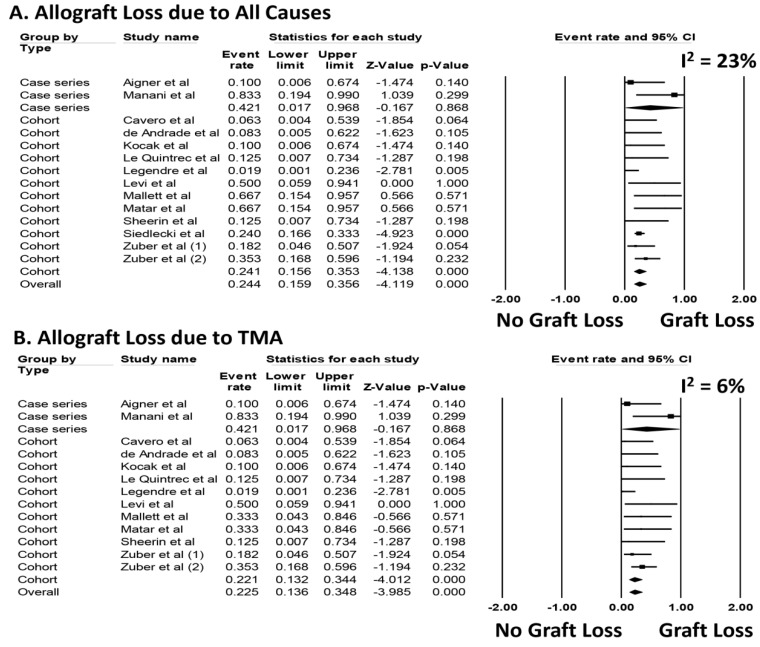
Incidence of (**A**) allograft loss due to all causes and (**B**) allograft loss due to TMA after kidney transplantation with therapeutic eculizumab.

**Figure 4 jcm-08-00919-f004:**
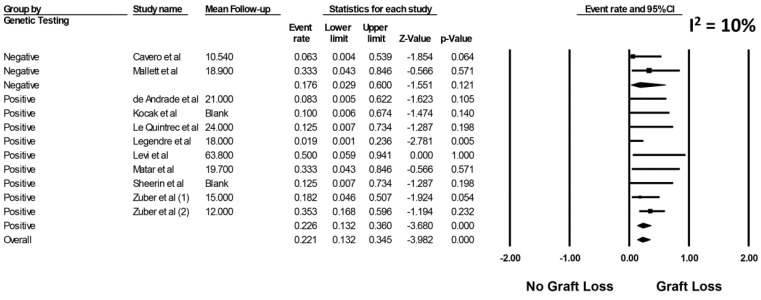
Forrest plot evaluating for the incidence of allograft loss due to TMA in cohort studies with data on genetic mutations.

**Figure 5 jcm-08-00919-f005:**
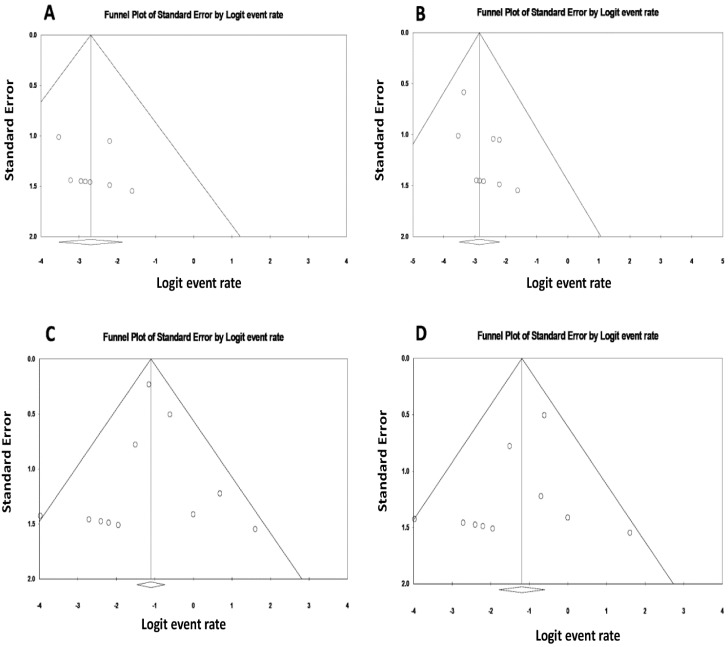
Funnel plots evaluating for publication bias for (**A**) the incidence of aHUS recurrence after kidney transplantation among patients who received prophylactic eculizumab; (**B**) the incidence of allograft loss due to TMA after kidney transplantation among patients who received prophylactic eculizumab; (**C**) the incidence of allograft loss due to all causes after kidney transplantation among patients with therapeutic eculizumab for aHUS recurrence; (**D**) the incidence of allograft loss due to TMA after kidney transplantation among patients with therapeutic eculizumab for aHUS recurrence.

**Table 1 jcm-08-00919-t001:** Use of eculizumab for treatment of atypical hemolytic uremic syndrome (aHUS) on patients after kidney transplantation.

Authors, Publication Year	Type of Study	Ktx Patients Treated with ecu, *n*	Median Age at Ktx, Months	Female, *n*	Patients with Prior Failed Ktx, *n*	Type of Donor	Ecu Initiation, Median Months	Duration of Therapy, Months	PLEX, %	Mean Follow up, Months	aHUS/ TMA Recurrence after Ecu, *n*	Graft Loss, *n*	Acute Rejection, *n*
Legendre et al., 2017	Cohort from non-randomized clinical trial	26	41.5	16	8	-	2	25	61.5	18	3	0	-
Levi et al., 2017	Case series	2	24.9	1	1	DD,1	1.25	55	0	63.8	1 ^a^	1	AMR, 1
Manani et al., 2017	Case series	2	42.5	1	0	LD, 1	0.5	1.5 ^b^	100	4		2	AMR,1
						DD, 1							CMR,1
Sheerin et al., 2016	Retrospective cohort	3	-	-	-	-	96 ^c^	Ecu cont.	0	-	3	0	-
Mallett et al., 2015	Retrospective cohort	3	40	3	0	DD, 2	29 ^d^	Ecu cont.	100	18.9	NM	2	AMR,1
						LR, 1							
Matar et al., 2014	Retrospective Cohort	3	38	2	2	LU,1	-	14.5 ^e^	100	19.7	2	2	0
						DD, 1							
						LR, 1							
LeQuintrec et al., 2013	Retrospective multicenter cohort	3	38	-	3	-	3	-	100	24	0	0	0
Zuber et al., 2012	Retrospective multicenter cohort	11	34	-	9	-	-	16	90	15	4	2	AMR, 1
Kocak, et al., 2015	Retrospective cohort	4	30.5	3	0	LR,1	2	-	100	-	-	0	-
Modelli de Andrade et al., 2017	Prospective cohort	5	32	2	0	DD, 4	1.16	21	-	21	0	0	0
						LD, 1							
Cavero et al., 2017	Retrospective multicenter cohort	7	43	5	-	-	0.46	2	85.7	10.54	-	0	0
Zuber et al., 2017	Retrospective multicenter cohort	17	-	-	-	-	-	12	0	12	-	6	-
Siedlecki et al., 2019	Retrospective multicenter cohort	100	39.5	31	3	-	40	24	-	24	-	24	-
Aigner et al., 2018	Case series	4	28.5	4	0	-	-	-	100	72	0	0	-

(-) missing information from the original study. a. One patient had recurrence after discontinuation followed by graft loss. b. One patient continued with therapy at the time of Manani et al. 2017 publication. c. One patient was started at 24 h post-transplant. d. One patient was started early after kidney transplantation. e. One patient continued with therapy at the time of Matar et al. 2014 publication. Abbreviations: Ktx: Kidney transplantation; *n*: Number; Ecu: Eculizumab; cont.: Continued; DD: Deceased donor; LD: Living donor; LU: Living unrelated donor; LR: Living related donor; AMR: Antibody mediated rejection; CMR: Chronic antibody mediated rejection; PLEX: Plasma exchange; TMA: Thrombotic microangiopathy.

**Table 2 jcm-08-00919-t002:** Use of eculizumab for prevention of aHUS on patients after kidney transplantation.

Authors, Publication Year	Type of Study	Ktx Patients Received prophylaxis with Ecu, *n*	Median Age at Ktx, Months	Female, *n*	Patients with Prior Failed Ktx, *n*	Type of Donor	Ecu Initiation, Median Days (D)	Duration of Therapy, Months	PLEX, %	Mean Follow up, Months	aHUS/ TMA Recurrence after Ecu, *n*	Graft Loss, *n*	Acute Rejection, *n*
Levi et al., 2017	Case series	10	41	5	4	DD, 7	D 0	Ecu cont. ^a^	0	21	1 ^b^	1	AMR, 2
						LD, 3							
Sheerin et al., 2015	Retrospective Cohort	8	-	-	-	-	-	-	-	-	0	0	-
Matar et al. 2014	Retrospective Cohort	4	39	3	3	LD, 1	D −1	6 ^c^	0	20.5	0	0	0
Zuber et al., 2012	Retrospective multicenter cohort	9	25.6	-	-	DD, 3	D 5, 1	Ecu cont.	66	14.5	0	1 ^d^	AMR +
							D 0, 2						ACR, 1
Modelli de Andrade et al., 2017	Prospective cohort	2	29	1	-	DD, 1	D 0	Ecu cont.	0	4	0	1 ^e^	-
Zuber et al., 2017	Multicenter retrospective cohort	35	-	63	-	-	D 0	-	0	35	17	1	-
Bresin et al., 2013	Case series from 4 independent multinational cohorts	4	-	-	-	-	-	-	-	-	0	0	-
Kumar et al., 2016	Case series	9	-	8	-	LU, 7	-	-	-	31.2	0	0	0
						DD, 2							
Yelken et al. 2017	Retrospective single center cohort	7	-	-	-	-	-	-	-	28	0	0	
Favi et al., 2017	Retrospective single center cohort	12	-	-	-	-	-	-	-	26.5	0	1	3
	Comparison group with no Ecu	24	-	-	-	-	-	-	-	79	11	7	7
Siedlecki et al. 2019	Multicenter cohort	88	32.3	45	24	-	-	Ecu cont.	-	27	-	3	-

(-) missing information from the original study. a. One patient was discontinued at 28.7 m with no complement mutation identified. b. One patient had recurrence after discontinuation followed by graft loss. There were subclinical TMA lesions on kidney biopsy in 2 patients. c. One patient continued prophylaxis at the time of Matar et al. 2014 publication. d. Patient had immediate graft loss due to intestinal hemorrhage. e. Patient had intestinal hemorrhage at 4 m after kidney transplantation. Abbreviations: Ktx: Kidney transplantation; *n*: Number; Ecu: Eculizumab cont.: Continued; D: Day; DD: Deceased donor; LD: Living donor; LU: Living unrelated donor; AMR: Antibody mediated rejection; ACR: Acute cellular rejection.

**Table 3 jcm-08-00919-t003:** Baseline characteristics of kidney transplant patients included in the meta-analysis *.

Baseline Characteristics	Treatment Group	Prophylaxis Group
Patients received eculizumab *n*/total (%)	192	188
Age at time of transplant, years (range)	38 (18–69)	32.3 (18–51)
Female, *n*/total (%)	94/158 (59.4)	125/148 (84.4)
History of previous transplant, *n*/total (%)	26/165 (15.8)	31/102 (30.4)
Type of donor, *n* (%)		
Living	8 (4.2)	11 (5.9)
Deceased	9 (4.7)	13 (6.9)
Not mentioned	175 (91.1)	164 (87.2)
Gene mutation, *n* (%)		
CFH	45 (23.4)	39 (20.7)
CFI	11 (5.7)	9 (4.8)
CFH/CFI	4 (2.0)	1 (0.5)
C3	10 (5.2)	10 (5.3)
CFHR1	1 (0.5)	-
CFHR2	2 (1)	-
CFHR3	1 (0.5)	-
CFH/CFHR3	1 (0.5)	-
CFH/CFHR1	2 (1)	-
CFHR1/CHFR3	2 (1)	-
CFH/CFI/CFHR3	1 (0.5)	-
CFI/C3	1 (0.5)	-
CFB	1 (0.5)	1 (0.5)
Anti CFH antibodies	1 (0.5)	3 (1.6)
MCP	3 (1.5)	9 (4.8)
MCP/THBD	1 (0.5)	-
THBD	1 (0.5)	-
No mutation identified	22 (11.5)	5 (2.6)
Not mentioned/not specified	82 (42.7)	111 (59)
Initiation of Eculizumab	2 months (median)	One day prior to surgery, *n*(%) 1 (0.5)
		On day of surgery, *n*(%) 57 (30.3)
		On day 5 post-surgery, *n*(%) 1 (0.5)
		Timing not mentioned, *n*(%) 129 (68.6)
Median follow up, months (range)	18.9 (3.9–72)	26.7 (4–79)
Plasmapheresis, *n*/total (%)	68/165 (41.2)	22/95 (23.1)

* Percentages are calculated from total available data. *n* = number.

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
