# Peer review of "Outcomes of Kidney Transplant Patients with Atypical Hemolytic Uremic Syndrome Treated with Eculizumab: A Systematic Review and Meta-Analysis"

_jcm, 2019, doi:10.3390/jcm8070919_

Reviewer 1 Report

This manuscript by Maria Jourdes Gonzalez Suarez et al. is a systematic review and meta-analysis focused on the outcomes observed with use of eculizumab for prevention and treatment of a HUS recurrence in kidney transplantation. They showed a possible advantageous role for anti-C5 antibody therapy in the prevention and management of recurrent aHUS. This is largely a sound paper although I have several concerns.

 1. Duineveld C et al. suggested that living donor kidney transplantation in aHUS without prophylactic eculizumab treatment appears feasible. This article has not been cited. Why had it been excluded? (Duineveld C, et al. Am J Kidney Dis. 2017 Dec;70(6):770-777. doi: 10.1053/j.ajkd.2017.06.024.)

 2. Description about eculizumab seems to be not enough in “Introduction” section. I suggest that more details such as adverse events and pharmacology and so on about eculizumab are described.

 3. In supplement table 1, there was a mistake about PLEX % in the literature by Aigner et al. in 2018.

 4. This study consists of 18 reports (13 cohort studies and 5 case series). However, 14 cohort studies and 4 case series were shown in table 1 and 2.

 Author Response

Reviewer #1
This manuscript by Maria Lourdes Gonzalez Suarez et al. is a systematic review and meta-analysis focused on the outcomes observed with use of eculizumab for prevention and treatment of a HUS recurrence in kidney transplantation. They showed a possible advantageous role for anti-C5 antibody therapy in the prevention and management of recurrent aHUS. This is largely a sound paper although I have several concerns.

 Response: We thank you for reviewing our manuscript and for your critical evaluation. We really appreciated your input and found your suggestions very helpful.

 Comment#1  Duineveld C et al. suggested that living donor kidney transplantation in aHUS without prophylactic eculizumab treatment appears feasible. This article has not been cited. Why had it been excluded? (Duineveld C, et al. Am J Kidney Dis. 2017 Dec;70(6):770-777. doi: 10.1053/j.ajkd.2017.06.024.)

 Response: We greatly appreciated the reviewer’s important input on this important study. We have cited this study as reference number #8. This study did not evaluate the use of eculizumab and thus is not included in the meta-analysis. However, we completely agree with the reviewer regarding the importance of this study’s result. We thus added this important point that Duineveld C et al. suggested that living donor kidney transplantation in aHUS without prophylactic eculizumab treatment appears feasible in the discussion. The following text in bold and the reference has been added in the revised manuscript as the reviewer’s suggestion.

 â€śWe acknowledge the limitations inherent in this study. Firstly, this systematic review included only observational studies (cohorts and case series). Consequently, the majority of available studies also lacked a comparison (control) group. Only one of the included studies had a comparison group who did not receive eculizumab [50]. Recently, Duineveld C et al. described 17 patients with aHUS who underwent living donor kidney transplantation without prophylactic eculizumab with median follow-up of 25 months after kidney transplantation [8]. Their institution’s protocol emphasized lower target tacrolimus level and blood pressure control among these patients with aHUS. They found only one patient developed aHUS recurrence at 68 days after kidney transplantation, and was treated successfully with eculizumab. The investigators suggested that living donor kidney transplantation in aHUS without prophylactic eculizumab treatment appears feasible. [8].”

  Comment#2 Description about eculizumab seems to be not enough in “Introduction” section. I suggest that more details such as adverse events and pharmacology and so on about eculizumab are described.

Response: We agree with the reviewer and found this suggestion helpful to improve our manuscript. We have added details on eculizumab on pharmacology and adverse events in the introduction to strengthen our manuscript as the reviewer’s suggestion.The following text in bold has been added in the revised manuscript as the reviewer’s suggestion.

 â€śTherapies described for management of aHUS in the post-transplant period include use of plasma exchange (PLEX), rituximab (used for anti-Factor H autoantibodies; helps to maintain low levels of antibodies, preventing recurrence of aHUS after transplant)[10], simultaneous liver-kidney transplant for CFH mutations, and use of eculizumab (a humanized monoclonal antibody directed against complement protein C5 and thus inhibits terminal complement activation) [11-13].  Prior to the use of eculizumab,  patients with gene mutations CFH, CFH-CFHR1/3, CFI, C3, and CFB had a 50% risk of progression to ESRD or death at onset of recurrent aHUS during the first year, and this risk increased to 75% after 3-5 years [14].

 â€śThe KDIGO workgroup recommends the prophylactic use of eculizumab in kidney transplant patients at high risk of recurrence based on their genetic mutations [14].  Whether there is an advantage of preemptive use of eculizumab in all patients with a known pre-transplant history of aHUS is currently unclear. In addition, eculizumab use is associated with an increased risk of infection due to terminal complement blockade such as meningococcal infections [15]. In this study, we aimed to assess the use of eculizumab in the prevention and treatment of aHUS recurrence after kidney transplantation.”

Comment#3 In supplement table 1, there was a mistake about PLEX % in the literature by Aigner et al. in 2018.

Response: We appreciated the reviewer’s thorough review. We apologize for this error. PLEX % in the literature by Aigner et al. 2018, should be 100% (4/4). We have made correction as the reviewer’s suggestion in the table. We also updated our analysis based on updated PLEX data as well. The following text has in bold has been corrected in the result of manuscript.

 â€śThe pooled estimated rates of allograft loss due to TMA was 22.5% (95%CI: 13.6-34.8%, I2 =6%, Figure 3B). Meta-regression analysis demonstrated no significant correlations between concomitant use of PLEX and rates of allograft loss due to all causes or due to TMA (P = 0.61 and 0.18, respectively).”

Comment#4 This study consists of 18 reports (13 cohort studies and 5 case series). However, 14 cohort studies and 4 case series were shown in table 1 and 2.   

Response: We appreciated the reviewer’s high quality and thorough review. We apologize for this error in Table 2. Bresin et al. 2013 described 4 case series. We have made correction in our Table 2. Once again, we greatly appreciated the reviewer’s thorough review.

All authors thank the Editors and reviewers for their valuable suggestions. The manuscript has been improved considerably by the suggested revisions! 

Reviewer 2 Report

Meat-analysis consists of  several publications including  small number of renal transplant recipients  treated with eculizumab due to aHUS recurrence or for prophylaxis. This meta-analysis includes large group of patients from aHUS registry. It is not clear if patients included in reported  small studies ale have been reported to aHUS registry (the same patients?). The presented group is heterogenous associated with difficulties in differentiation between  primary (aHUS complement dependent) and secondary HUS. Results my be dependent on diagnostic test, therapy schemes, duration, further PLEX. The beneficial effect of eculizumab in aHUS is well known from literature. The results of presented analysis do not provide any additional knowledge especially in the light of published results by Siedlecki et al of aHUS registry.

Author Response

Reviewer #2
Meta-analysis consists of several publications including small number of renal transplant recipients treated with eculizumab due to aHUS recurrence or for prophylaxis.

Response: We thank you for reviewing our manuscript and for your critical evaluation. We really appreciated your input and found your suggestions very helpful.

 Comment#1 

This meta-analysis includes large group of patients from aHUS registry. It is not clear if patients included in reported small studies ale have been reported to aHUS registry (the same patients?).

Response: The reviewer made a very good point. We thus reviewed all the literature and all included studies again. We additionally performed sensitivity analysis excluding studies with potentially duplicated patients. The following text and results in bold have been added to the revised manuscript as the reviewer suggestion.

“Sensitivity analysis excluding studies with potentially duplicated patients was performed, and showed the pooled estimated rates of aHUS recurrence and allograft loss due to TMA of 5.5% (95%CI: 1.9-14.6%, I2 =0%, Figure S1) and 5.3% (95%CI: 2.4-11.3%, I2 =0%, Figure S2), respectively.”

“Sensitivity analysis excluding studies that potentially included duplicate patients was performed, and showed the pooled estimated rates of allograft loss due to all causes was 26.7% (95%CI: 13.0-46.9%, I2 =33%, Figure S3) and the pooled estimated rates of allograft loss due to TMA was 20.0% (95%CI: 9.1-38.4%, I2= 29%, Figure S4), respectively.”

Comment#2  

The presented group is heterogeneous associated with difficulties in differentiation between primary (aHUS complement dependent) and secondary HUS. Results may be dependent on diagnostic test, therapy schemes, duration, further PLEX.

Response:  We appreciated the reviewer’s helpful comments. Thus, we additionally collected data on PLEX in Table 1-2. We also additionally performed analysis based on genetic mutations associated with aHUS as the reviewer’s suggestion. The following text and results in bold have been added to the revised manuscript as the reviewer suggestion.

 â€śMeta-regression analysis demonstrated no significant correlations between concomitant use of PLEX and rates of allograft loss due to all causes or due to TMA (P = 0.650.61 and 0.460.18, respectively).”

“The pooled estimated incidence of allograft loss due to TMA in cohort studies with data on genetic mutations was 22.6% (95%CI: 13.2-36.0%, I2 =10% Figure 4).” 

Comment#3

The beneficial effect of eculizumab in aHUS is well known from literature. The results of presented analysis do not provide any additional knowledge especially in the light of published results by Siedlecki et al of aHUS registry.

Response: We appreciated the reviewer’s important input. We agree with your comment on the potentially beneficial effect of eculizumab in aHUS. However, there are inconsistency in the use of eculizumab for aHUS patients undergoing kidney transplantation in the U.S. and worldwide. Very recently, Duineveld C et al. described 17 patients with aHUS who underwent living donor kidney transplantation without prophylactic eculizumab with median follow-up of 25 months after kidney transplantation. They suggested that living donor kidney transplantation in aHUS without prophylactic eculizumab treatment appears feasible. Given the inconsistency and limited data on this topic, we are hoping that the findings from our study will help encourage future research in this area. In addition, our systematic review also included patients that are not included in the country of aHUS registry in a great study by Siedlecki et al 2019.

We respect the reviewer’s helpful comments and thus provided additional sensitivity analyses and meta-regression analysis as mentioned above. We also added additional limitations in our discussion. The following text in bold has been added in bold in the discussion/limitation section:

“We acknowledge the limitations inherent in this study. Firstly, this systematic review included only observational studies (cohorts and case series). Consequently, the majority of available studies also lacked a comparison (control) group. Only one of the included studies had a comparison group who did not receive eculizumab [54]. Recently, Duineveld C et al. described 17 patients with aHUS who underwent living donor kidney transplantation without prophylactic eculizumab with median follow-up of 25 months after kidney transplantation [8]. Their institution’s protocol emphasized lower target tacrolimus level and blood pressure control among these patients with aHUS. They found only one patient developed aHUS recurrence at 68 days after kidney transplantation, and was treated successfully with eculizumab. The investigators suggested that living donor kidney transplantation in aHUS without prophylactic eculizumab treatment appears feasible. [8]. Secondly, inconsistencies in the reporting of certain variables such as type of donor, history of previous transplantation, genetic mutations, and immunosuppression regimen can make it difficult to draw firm associations between eculizumab prophylaxis or treatment on renal graft outcomes. Caution should be exercised in interpreting these data. Lastly, this study analysis was conducted on a highly selected study population. Only studies in which eculizumab was used were analyzed, and individuals who received an alternate treatment course or did not receive eculizumab were excluded from our analysis. Therefore, the frequency of mutations might not be representative. Although funnel plots and Egger's test of event rates demonstrated no statistical significance, a low risk of publication bias cannot be implied given the lack of comparison groups. Thus, future studies to assess the responsiveness to eculizumab based on certain aHUS genetic mutations (low risk vs moderate risk vs high risk of recurrence)[6] are needed.

All authors thank the Editors and reviewers for their valuable suggestions. The manuscript has been improved considerably by the suggested revisions! 

Round  2

Reviewer 2 Report

I accept all your  responses and improvements of  the manuscript.